# INRSTEG: FLEXIBLE CROSS-MODAL LARGE CAPACITY STEGANOGRAPHY VIA IMPLICIT REPRESENTATIONS

## ABSTRACT

We present INRSteg, an innovative lossless steganography framework based on a novel data form Implicit Neural Representations (INR) that is modal-agnostic. Our framework is considered for effectively hiding multiple data without altering the original INR ensuring high-quality stego data. The neural representations of secret data are first concatenated to have independent paths that do not overlap, then weight freezing techniques are applied to the diagonal blocks of the weight matrices for the concatenated network to preserve the weights of secret data while additional free weights in the off-diagonal blocks of weight matrices are fitted to the cover data. Our framework can perform unexplored cross-modal steganography for various modalities including image, audio, video, and 3D shapes, and it achieves state-of-the-art performance compared to previous intra-modal steganographic methods.

## 1 INTRODUCTION

Nowadays, an unprecedented volume of data is shared and stored across various digital platforms and the volume is bound to intensify even further in the future. This proliferation and accessibility of data contributes to convenience in our lives and enables innovation and advancement that weren't achievable in the past; however, it also brings a concerning challenge to the security of information. Steganography, derived from the Greek words "steganos" (covered) and "graphein" (writing), aims to hide the secret information inside a cover data resulting in stego data, also called the container. Unlike cryptography, which investigates in hiding the data of interest in a coded form, steganography operates with imperceptible concealment so that the existence of the secret data is undetectable.

Up to now, image is the most commonly used modality of cover data, as it is less sensitive to human perception and is comparatively simple to embed secret information. However, image tends to have a limit on payload capacity, and is susceptible to detection. Deep learning methods (Baluja, 2017; Zhu et al., 2018) have increased the security, but suffer from unstable extraction and high computational costs (Subramanian et al., 2021). Other modalities such as audio, video and 3D shape also suffer from similar limitations (Liu et al., 2019; Zhou et al., 2022) and undergo trade-offs between capacity and security. Moreover, for cross-modal steganography, where the modality of the secret and the cover data differs (Huu et al., 2019; Han et al., 2023; Yang et al., 2019), these limitations intensify due to the incompatibility between different data modalities.

Implicit neural representation (INR) has recently attracted extensive attention as an alternative data representation, using continuous and differentiable functions to encode or process data via parameters of a neural network. The concept of INR offers a more flexible and expressive approach than the conventional notion of discrete representations. Multiple modalities, as audio, image, video and 3d shape, are all presentable using a single neural network in various resolutions (Dupont et al., 2022). Additionally, INR benefits in capacity by reducing the memory cost according to the network design (Lee et al., 2021; Park et al., 2019).

In this paper, we present INRSteg, the first cross-modal steganography framework available across a diverse range of modalities, including images, audio, video, and 3D shapes. All data, including the secret data and the stego data, are expressed in the form of INRs, allowing to take advantage of the flexibility and high capacity inherent in INRs. This flexibility to all modalities also enable cross-modal tasks that remain largely unexplored due to the difficulties that emerge when the dimension or capacity of the secret data exceed that of the cover data or when the secret data contains temporal

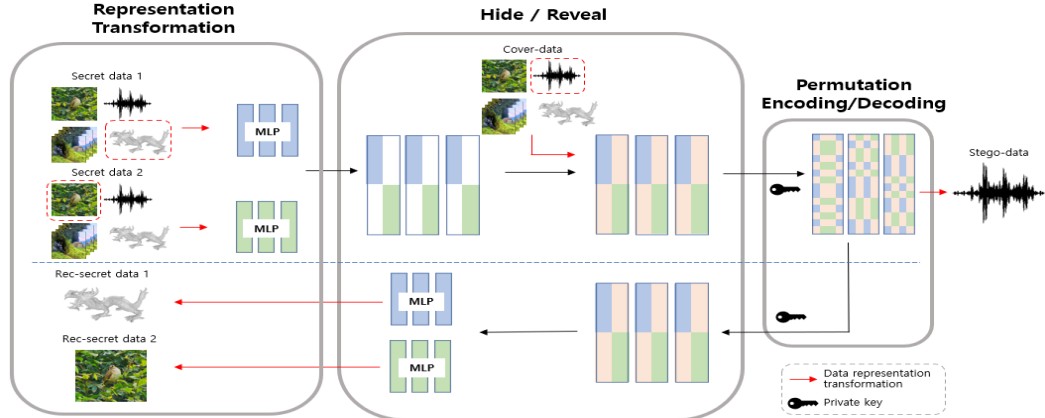

Figure 1: A general framework of INRSteg for hiding two types of secret data. After the representation transformation of each secret data, the weights of two INRs are concatenated to form a new MLP network. With the weight freezing technique, the weights from secret data are fixed while the rest are fitted on cover data. The existence of secret data is concealed by permutation encoding via private key, which does not affect the reconstruction performance. The private key is then used to decode the permuted network, and the secret data are revealed by separating each MLP network.

information. Along with the versatility across the modalities, our framework further enables multiple secret data into one cover data without loss of information as shown in Figure 1. The hiding process first converts all secret data into INRs. When there is only one secret data, the network is padded with additional nodes and weights, and if not, the networks of multiple secret data are concatenated as illustrated in Figure 1. This concatenated network is then fitted on the cover data, resulting in a stego INR, which appears identical to the cover data when reconstructed. To enhance the security, permutation using a private key is additionally executed. Revealing the secret data is executed by re-permuting and disassembling the weights of the stego INR. This procedure is not only simple but also guarantees a lossless revealing process as all neural representations of the secret data are fully recoverable. Also, by not requiring any deep learning models to be trained for the hiding and revealing process, such as GAN (Shi et al., 2018; Yang et al., 2018), and thus excluding the need for any training data, our framework prevents biases from training datasets (Mehrabi et al., 2021) and domain shift issues, allowing datasets of all modalities to be applied without any additional pre-processing.

The proposed steganography framework referred to as INRSteg outperforms prior frameworks such as DeepMIH (Guan et al., 2023) and DeepSteg (Baluja, 2017), and with INRSteg, it is possible to perform unexplored cross-modal steganography such that it is possible to hide 3D shapes in audio. These are our main contributions, and the details will be described in further sections:

- Introduce an Implicit Neural Representation (INR) steganography framework that flexibly adopts to a large range of modalities and further enable all cross-modal steganography.

- Demonstrate a neural network stacking procedure that enables a lossless hiding and revealing of multiple secret data at once.

- A comprehensive evaluation on the steganography metrics that shows our work achieves state-of-the-art cross-modal steganography.

## 2 RELATED WORK

### 2.1 STEGANOGRAPHY

Steganography is a technique that focuses on hiding confidential information by adeptly embedding data within other publicly accessible data, effectively concealing the very existence of the hidden message. When one is aware of this concealed data, it can be retrieved by a revealing process with an appropriate stego key. The foundational framework of steganography was defined in Simmons' work(Simmons, 1985), where the dynamics between steganography and steganalysis came from a 3-player game. A prominent approach in historical methods is

the Least Significant Bit (LSB) based method, which takes advantage of the imperceptibility of small changes to the least significant bits of the data(Jain & Ahirwal, 2010; Zhang et al., 2009). With the rise of Deep neural networks, Generative Adversarial Networks (GANs) are leveraged to hide binary data in steganographic images significantly improving capacity(Zhang et al., 2019; Tan et al., 2022). Recently, LISO proposes a novel optimization-based steganography algorithm, which is fast and recovers better quality secret images(Chen et al., 2022). Notwithstanding their innovations, a recurrent limitation across these previ-

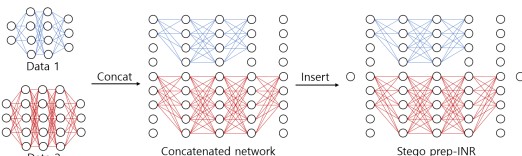

Figure 2: It shows an example of concatenating two secret INRs and network preparation for stego INR. For this case, $I_1 = 2, O_1 = 3, I_2 = 3, O_2 = 3, n_1 = 2, n_2 = 3, D_1 = 4, D_2 = 5, I_{cover} = 1, O_{cover} = 1$. Note that the free weights and biases are omitted for clarity.

ous methods is their restricted data embedding capacity. If the cover data is damaged in any way during the embedding process, it becomes susceptible to detection by steganalysis algorithms(You et al., 2021; Xu et al., 2016). This vulnerability poses a significant limitation, especially when attempting to hide large volumes of data such as videos or 3d shapes.

## 2.2 CROSS-MODAL STEGANOGRAPHY

The evolution of steganography has also expanded to the concealment of information across different data modalities, such as embedding audio data within images or videos, which is called cross-modal steganography. (Yang et al., 2019) embeds video information into audio signals by leveraging the human visual system's inability to recognize high-frequency flickering, allowing for the hidden video to be decoded by recording the sound from a speaker and processing it. (Huu et al., 2019) utilizes a joint deep neural network architecture comprising two sub-models, one for embedding the audio into an image and another for decoding the image to retrieve the original audio. These methods achieve cross-modal steganography using deep learning techniques, but are still restricted on specified modalities.

Recent advancements in Implicit Neural Representations (INRs) have paved the way for a new paradigm in steganography(Li et al., 2023; Han et al., 2023). INRs, with their ability to represent multiple types of data into the unified structures, have been leveraged for representing multiple modalities of data simultaneously. (Li et al., 2023) presents StegaNeRF, a method to hide images at viewpoints when rendering NeRF, showing not only images but also other modality data can be hidden. (Han et al., 2023) proposes an innovative method to conceal audio, video, and 3D shape data within a cover image, leveraging RGB values as weights of INRs. This method involves transmitting a stego image along with the remaining networks, typically the front and back weights, equipped with INR structural information. Subsequently, the concealed message can be retrieved through the forward processing of the INR network. While these recent works represent a paradigm shift in steganographic techniques, a persistent challenge has been the trade-off between the reconstruction error of the hidden message and the distortion error of the stego image. Our proposed method in this paper seeks to address and overcome this limitation while maintaining the advantages of INRs, which demonstrates a significant advancement in steganography research.

## 3 METHOD

### 3.1 FRAMEWORK

Our research focuses on INR steganography, where the data representation is transformed into Implicit Neural Representations (INR), such as SIREN Sitzmann et al. (2020). Figure 1 shows the general framework of our methodology, which is segmented into three sub-stages. Firstly, secret data, denoted as $x_{secret}$, are transmuted into its corresponding Implicit Neural Representations, represented as $\theta_{secret}$. Subsequently, in the second stage, these secret INRs, $\theta_{secret}$, are concealed within a new INR, $\theta_{stego}$, which is fitted to represent $x_{cover}$ while $x_{stego}$ is the reconstructed from of $\theta_{stego}$. The final stage introduces a strategic layer-permutation mechanism, designed to avoid detection of the secret data's presence and location. We summarize the main notations with explanation in Appendix A and provide pseudo-code algorithms in Appendix B.

## 3.2 Data Representation and Transformation

In the first stage, we represent secret data as Implicit Neural Representations (INRs) by fitting INRs to each data. The structure of an INR is a Multi-Layer Perceptron (MLP) network, which employs activation function $\sigma(.)$ with $n$ hidden layers of size $D$, and the outputs can be represented as follows:

$$\mathbf{y} = \mathbf{W}^{(n)} \left( g_{n-1} \circ \cdots \circ g_1 \circ g_0 \right) (\mathbf{x}_0) + \mathbf{b}^{(n)}, \text{where } \mathbf{x}_{i+1} = g_i\left(\mathbf{x}_i\right) = \sigma\left(\mathbf{W}^{(i)}\mathbf{x}_i + \mathbf{b}^{(i)}\right). \quad (1)$$

where $\mathbf{x_i}$ is the input of the $i^{\text{th}}$ layer for $i \in \{0, 1, 2, ..., n\}$, $\mathbf{y} \in \mathbb{R}^O$ is the output value corresponding to $\mathbf{x_0} \in \mathbb{R}^I$, and $g_i : \mathbb{R}^D \rightarrow \mathbb{R}^D$. We use SIREN (Sinusoidal Representations for Neural Networks) Sitzmann et al. (2020) as the base structure of the INR network, which uses the sinusoidal activation function, since it is widely used in the INR research field. In general, given a set $S$ of inputs $\mathbf{x} \in \mathbb{R}^I$ and the corresponding outputs $\mathbf{y} \in \mathbb{R}^O$, the loss function is given by

$$\mathcal{L}_{recon}(\theta) = \sum_{(\mathbf{x},\mathbf{y}) \in \mathcal{S}} \|f_\theta\left(\mathbf{x}\right) - \mathbf{y}\|_2^2 \quad (2)$$

where $f_\theta$ is the neural representation of the data. For example, given a 2D image, the set $\mathcal{S}$ consists of 2D coordinates $\mathbf{x} \in \mathbb{R}^2$ and the corresponding RGB values $\mathbf{y} \in \mathbb{R}^3$. For our experiments, input and output dimensions $(I, O)$ are $(1, 1)$, $(2, 3)$, $(3, 1)$, $(3, 3)$ for audio, image, 3D shapes, and video, respectively. INR enables a variety of data types to be expressed in a unified architecture. This benefits the overall security paradigm of our steganography framework by hiding the underlying data types.

## 3.3 Hiding Stage

In this section, we explain our main idea, which is to hide the secret data within the cover data during the second stage, where the secret INRs, denoted as $\theta_{secret}$, undergo a process of concatenation and fitting to the cover data, $x_{cover}$. Within the weight space, only the diagonal blocks of weight matrices are occupied by the secret data, so that the non-diagonal blocks can be fitted to the cover data, $x_{cover}$, while ensuring the diagonal block's weight values to remain unaltered.

First, we explain when the number of secret data $N = 2$ without losing generality for hiding multiple data. The two secret data do not have to belong to the same modality. As the structure of the secret INRs can differ, the number of input nodes, output nodes, hidden layers, and the dimension of the hidden layers are each set to $(I_1, O_1, n_1, D_1)$, $(I_2, O_2, n_2, D_2)$, respectively. And $(I, O, n + 2, D_1 + D_2)$ for the stego data. Now, we integrate these two secret networks into a single network by concatenating them as shown in Figure 2. For the base case when $n_1 = n_2$, the parameters of the concatenated network are as follows:

$$\mathbf{W}^{(i)} = \begin{pmatrix} \mathbf{W}_1^{(i)} & \mathbf{W}_{01}^{(i)} \\ \mathbf{W}_{02}^{(i)} & \mathbf{W}_2^{(i)} \end{pmatrix}, \mathbf{b}^{(i)} = \begin{pmatrix} \mathbf{b}_1^{(i)} \\ \mathbf{b}_2^{(i)} \end{pmatrix} \quad (3)$$

Here, $n = n_1 = n_2$, and $\mathbf{W}_1^{(i)}$, $\mathbf{W}_2^{(i)}$ and $\mathbf{b}_1^{(i)}$, $\mathbf{b}_2^{(i)}$ are the $i^{\text{th}}$ layer's weight matrices and bias vectors of the two secret INRs being mixed, where $i \in \{0, ..., n\}$. For $i = 0$ and $i = n$, the weight matrices are expanded to a square matrix by padding them with free parameters. $\mathbf{W}_{01}^{(i)}$ and $\mathbf{W}_{02}^{(i)}$ are matrices with additional free parameters.

Next, we explore when $n_1 < n_2 = n$. For $n_1 < n_2$, as shown in Figure 2, we start concatenating from the front layer in the same way as the base case by placing the weights in each diagonal block. The secret INR $\theta_1$ has all of its weights placed first. As there are no more weight matrices to concatenate with $\theta_2$, we use matrices with free parameters as $\mathbf{W}_1^{(i)}$ for $i > n_1$ so the concatenation can continue with $\theta_2$. Bias is also handled in a similar way.

After concatenating the two secret INRs, a stego INR is prepared by adding an input layer with $I$ nodes and an output layer with $O$ nodes to the front and back of the concatenated network, respectively. The resulting weight matrix dimension of the first hidden layer becomes $((D_1 + D_2) \times I)$ and that of output layer becomes $(O \times (D_1 + D_2))$ as in Figure 2. Now, the stego INR is fitted with the cover data ensuring that the weights of the secret data in the concatenated INR are frozen. Note that the free parameters of the stego INR should be initialized before training for better performance.

After successfully fitting the stego INR $\theta_{stego}$, stego data $x_{stego}$ is revealed from the forward process of the stego INR and it becomes possible to retrieve all secret data from the $\theta_{stego}$.

For $N > 2$, the hiding process is expanded in the same manner, so we further explain how to hide when $N = 1$. When there is only one secret data, free parameters to fit the cover data can be generated by padding additional nodes and weights to the secret INR. Our method permits selective region allocation of the secret INR within the stego INR, followed by updates to the unoccupied regions to align with the cover data. Suppose that the secret data is located in the middle of the stego INR, then the formula is as follows.

$$\mathbf{W}^{(i)} = \left( \begin{array}{ccc} \mathbf{W}_{01}^{(i)} & \mathbf{W}_{02}^{(i)} & \mathbf{W}_{03}^{(i)} \\ \mathbf{W}_{04}^{(i)} & \mathbf{W}_{1}^{(i)} & \mathbf{W}_{05}^{(i)} \\ \mathbf{W}_{06}^{(i)} & \mathbf{W}_{07}^{(i)} & \mathbf{W}_{08}^{(i)} \end{array} \right), \mathbf{b}^{(i)} = \left( \begin{array}{c} \mathbf{b}_{01}^{(i)} \\ \mathbf{b}_{1}^{(i)} \\ \mathbf{b}_{02}^{(i)} \end{array} \right), \tag{4}$$

where $n = n_1$. $\mathbf{W}_1^{(i)}$ and $\mathbf{b}_1^{(i)}$ are the weight and bias of the $i^{\text{th}}$ layer, where $i \in \{0, 1, ..., n\}$, and $\mathbf{W}_{01}^{(i)}, \mathbf{W}_{02}^{(i)}, \cdots, \mathbf{W}_{08}^{(i)}$ are matrices with additional free parameters. This network is then considered as the concatenated network and goes through the same processes as when $N = 2$.

The inherent flexibility of our approach is highlighted by its capability to insert diverse data types, in varying amounts, at arbitrary desired positions. This versatility addresses and mitigates capacity constraints, exemplified by scenarios such as hiding multiple videos within a singular audio stream.

## 3.4 PERMUTATION ENCODING

To enhance security of our method, we strategically permute the nodes within the stego network, which allows the weights of the secret INRs to be distributed throughout the stego INR, making it visually undetectable. In this section, we explain how to permute the nodes of each layer using one 128-bit private key $\rho$, utilizing a cryptographic key derivation function (KDF), which derives one or more secret keys from a master key. First, the private key is randomly selected and is employed to generate $n$ secret keys for $n$ hidden layers. Subsequently, the nodes of each layer are permuted according to the secret key assigned to each layer. The total number of possible permutation variations per INR is denoted as $(\mathbf{D}!)^n$, where $D$ is the number of nodes of the hidden layer. This shows that the security implications of this permutation are amplified as the possible combinations of permutation increase exponentially according to the number of hidden layers of the stego INR.

It is crucial to emphasize that the overall functionality of the neural network remains unchanged throughout this operation. This operation leverages the inherent property of permutation invariance exhibited by Multi-Layer Perceptrons (MLPs) within a single layer. As INRs are also MLP networks, all permuted networks are identical to the original, so the stego network is invariant to the permutation, whereas extracting the subnetworks for secrete data becomes impossible now without the permutation information.

## 4 EXPERIMENT

The performance of INRSteg with regards to resilience to distortion, capacity and security is compared to that of current state-of-the-art methods. Distortion measures both the indistinguishability between the cover data and the stego data, and the performance drop during the hiding and revealing stage of the secret data. Capacity denotes the volume of hidden information that the cover data can accommodate. Security indicates the extent to which the secret data can avoid detection when subjected to steganalysis. For distortion, we measure several metrics for images, audio, and video, and visualization for 3D shapes. For security, we compute the detection accuracy using steganalysis models, SiaStegNet (You et al., 2021) and XuNet (Xu, 2017). In terms of capacity, we present multiple cross-modal steganography experiments.

### 4.1 EXPERIMENTAL SETTING

**Dataset Setup** For ImageNet (Deng et al., 2009), each image data is resized to 256 x 256 resolution. We further utilize mini-ImageNet for steganalysis experiments. For GTZAN music genre dataset(Tzanetakis & Cook, 2002), each audio data is cropped to a length of 100,000 samples with

a 22,050 sample rate. For the scikit-video dataset, each video data is resized to $128 \times 128$ resolution and 16 frames are extracted. From the Stanford 3D Scanning Repository, we select two samples, the Asian Dragon and the Armadillo.

**Implicit Neural Representation Setting** For INR, we use SIREN (Sitzmann et al., 2020) as the base structure. For steganography, we use 6 hidden layers for all secret data modalities. The size of the hidden layer is 128, 128, 192, and 192 for image, audio, video, and 3d shapes, respectively. The stego data consists of 8 hidden layers except for 3D shapes in the multi-data steganography task, where 9 hidden layers are used. For single-data steganography, the neural networks are padded with a ratio of 1.0. For steganalysis experiments using images, we use 5 hidden layers with a size of 128. The Adam optimizer with learning rate 1e-4 is used for all modalities. For ablation study, we conduct additional experiments to figure out the effectiveness of padding ratio in Appendix D.

**Distortion evaluation metric** To evaluate the distortion between cover and stego data, and secret and revealed secret data, we visualize 3D shapes and employ several evaluation metrics for image, audio, and video. For image data, we employ Peak Signal-to-Noise Ratio (PSNR) (Wang et al., 2004), Structural Similarity Index (SSIM), Mean Absolute Error (MAE), and Root Mean Square Error (RMSE). For video data, in addition to PSNR and SSIM we evaluate Average Pixel Discrepancy (APD) and Perceptual Similarity (LPIPS)(Zhang et al., 2018). For audio data, we employ MAE and Signal-to-Noise Ratio (SNR). All data are represented within the range of 0 to 255.

|       |          | Cover | | |
|-------|----------|-------|-------|-------|
|       |          | image | audio | video |
| Secret | image   | 62.34 / 0.204 | 32.63 / 0.599 | 35.26 / 3.315 |
|        | audio   | 63.16 / 0.179 | **41.42 / 0.268** | 34.69 / 3.574 |
|        | video   | 56.18 / 0.408 | 27.20 / 0.931 | **41.26 / 1.532** |
|        | 3D shape | **92.95 / 0.000** | 23.12 / 1.403 | 38.54 / 2.244 |

|       | Metric |
|-------|--------|
| image | 45.84 / 1.326 |
| audio | 25.70 / 0.854 |
| video | 36.64 / 2.907 |

Table 1: Cover/Stego performance of single-data cross-modal steganography. The evaluation metrics are as following: image (PSNR↑/RMSE↓), audio (SNR↑/MAE↓), and video (PSNR↑/APD↓). The best results are **highlighted** for each cover modality.

Table 2: Secret/Revealed secret performance of single-data cross-modal steganography.

| Metric | MAE/APD ↓ | SNR/PSNR ↑ | SSIM ↑ | LPIPS ↓ |
|--------|-----------|------------|--------|---------|
| cover (audio)    | 0.497 | 30.91 | -     | -     |
| secret 1 (audio) | 0.854 | 25.70 | -     | -     |
| secret 2 (video) | 2.907 | 36.64 | 0.997 | 0.032 |
| cover (video)    | 3.366 | 35.38 | 0.997 | 0.060 |
| secret 1 (image) | 1.084 | 45.84 | 0.988 | 1.326 |
| cover (image)    | 0.740 | 48.58 | 0.993 | 0.944 |
| secret 1 (video) | 2.907 | 36.64 | 0.997 | 0.032 |

Table 3: Performance of three steganography tasks: audio and video to audio, image and 3D shape to video, and video and 3D shape to image. Metrics for 3D shapes are excluded as they are visualized.

## 4.2 CROSS-MODAL STEGANOGRAPHY

We evaluate the distortion performance of all single-data cross-modal steganography tasks. As presented in Table 1 and Table 3, our framework achieves exceptional performance for cover/stego pairs in multi and all single-data cross-modal tasks. Specifically, the cover/stego performance improvement is affected by two components: the modality of the secret and cover data and the neural network capacity of the stego INR. When the modality is identical, the stego INR is inclined to fit more stable and accurately to the cover data. This is due to the inherent characteristic of INRs that the weight spaces of the neural networks tend to demonstrate similarity within the same modality Tancik et al. (2021). Also owing to the use of INRs, larger neural network in the cover data fitting process results in superior stego data reconstruction. This is shown in Table 1, where the performance of hiding 3D shapes in images outperforms hiding images in images. For the secret and the

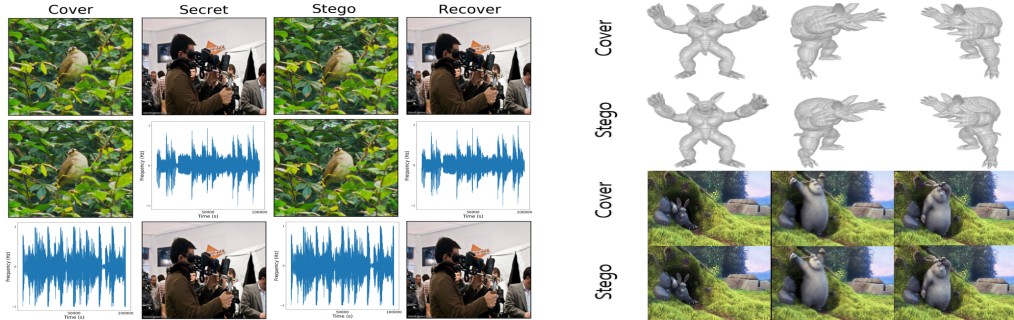

Figure 3: Image-to-image, audio-to-image, and image-to-audio steganography in order from top to the bottom.

Figure 4: Stego/Cover visualization for multi-data steganography: 3D shape and audio to 3D shape, and image and 3D shape to video.

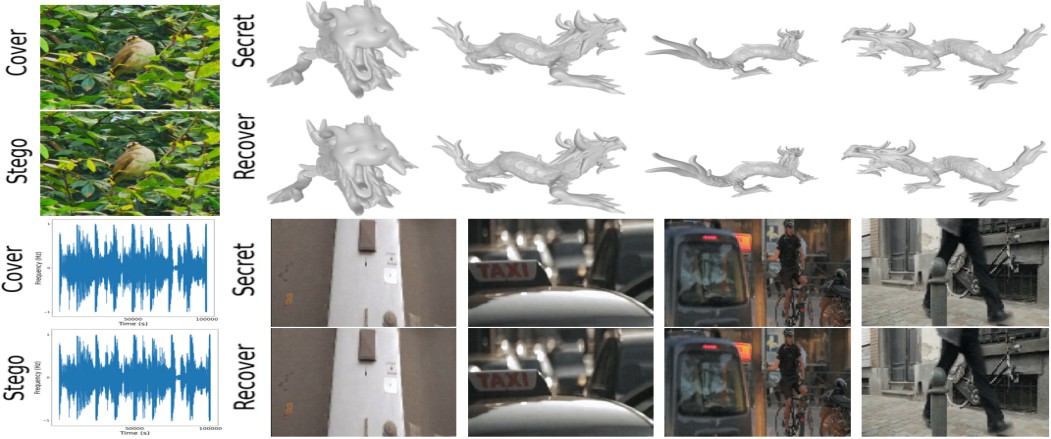

Figure 5: 3D shape-to-image and video-to-audio steganography in order from the top to the bottom.

revealed secret data pair, as our framework goes through lossless recovery in terms of INR, the secret/revealed secret performance is solely dependent on the data representation transformation phase is presented in Table 2. This performance can either be increased or decreased flexibly as one of the strengths of our framework is that we can control the secret/revealed secret performance by manipulating the network size for the secret data. By enlarging the network size, the performance continues to increase as the network complexity increases. On the other hand, the size of the INR network can be reduced to save resource space while maintaining acceptable recovery quality, and the relevant experimental results are detailed in next section. Figure 3 and Figure 5 shows reconstruction figures of various single-data steganography for intra and cross-modal cases. Additionally, Figure 4 represents cover/stego reconstruction figures for multi-data experiments showing successful multi-data steganography with almost no visible difference for both 3D shape and video. For tasks where 3D shapes are the cover data, the stego data may be unstable as the transformation phase of 3D shapes into INRs with SIREN Sitzmann et al. (2020), which is the base structure we employed for INR, can be unstable as further explained in Appendix C.

| | Metric | MAE ↓ | PSNR ↑ | SSIM ↑ | RMSE ↓ |
|---|---|---|---|---|---|
| DeepSteg | cover | 2.576 | 37.48 | 0.947 | 3.417 |
| | secret | 3.570 | 33.34 | 0.957 | 3.57 |
| DeepMIH | cover | 2.839 | 36.05 | 0.932 | 4.276 |
| | secret | 3.604 | 32.97 | 0.9550 | 5.746 |
| INRSteg | cover | **0.153** | **62.34** | **0.999** | **0.204** |
| | secret | **1.084** | **45.84** | **0.988** | **1.326** |

Table 4: Performance comparison of image-to-image steganography.

| | Accuracy (%) | |
|---|---|---|
| | SiaStegNet | XuNet |
| DeepSteg | 92.87 | 75.83 |
| DeepMIH | 90.70 | 90.82 |
| INRSteg | **50.00** | **50.00** |

Table 5: Steganalysis comparison of image-to-image steganography.

| | Metric | MAE ↓ | PSNR ↑ | SSIM ↑ | RMSE ↓ |
|---|---|---|---|---|---|
| DeepMIH | cover | 2.754 | 35.78 | 0.941 | 4.318 |
| | secret 1 | 3.604 | 32.97 | 0.9550 | 5.746 |
| | secret 2 | 2.6095 | 36.34 | 0.959 | 3.919 |
| INRSteg | cover | **0.204** | **62.42** | **0.999** | **0.255** |
| | secret 1, 2 | **1.084** | **45.84** | **0.988** | **1.326** |

| | Accuracy (%) | |
|---|---|---|
| | SiaStegNet | XuNet |
| DeepMIH | 87.96 | 68.52 |
| INRSteg | **50.00** | **50.00** |

Table 6: Performance comparison of two images into one image steganography.

Table 7: Steganalysis comparison of two images into one image steganography.

## 4.3 INTRA-MODAL STEGANOGRAPHY

We compare INRSteg to existing image-to-image steganography methods to show that our framework also excels in intra-modal tasks. Table 4 and Table 6 compares the performance of INRSteg with other deep learning based image-to-image steganography methods, DeepMIH Guan et al. (2023) and an improved DeepSteg Baluja (2017). INRSteg outperforms on all metrics showing that our framework achieves state-of-the-art performance. Figure 6 further visualizes the cover/stego and secret/revealed secret image pairs for two images into one steganography. We continue to compare our results with DeepMIH using difference maps each enhanced by 10, 20, and 30 times. For INRSteg, nearly no visual differences exist between the original and the revealed images even when the difference map is enhanced 30 times. For DeepMIH, on the other hand, we can notice obvious differences for all difference maps when comparing cover/stego and secret1,2/revealed secret 1,2 pairs.

| | Metric | MAE ↓ | PSNR ↑ | SSIM ↑ | RMSE ↓ |
|---|---|---|---|---|---|
| 3 images | cover | 0.153 | 63.26 | 1.000 | 0.204 |
| | secret 1,2,3 | 1.084 | 45.84 | 0.988 | 1.326 |
| 4 images | cover | 0.128 | 63.73 | 1.000 | 0.179 |
| | secret 1,2,3,4 | 1.084 | 45.84 | 0.988 | 1.326 |

Table 8: Distortion performance of hiding multiple images ($> 2$) into one.

| | Metric | PSNR ↑ | RMSE ↓ |
|---|---|---|---|
| 3 images | cover | 62.31 | 0.204 |
| | secret 1,2,3 | 43.67 | 1.642 |
| 4 images | cover | 62.58 | 0.153 |
| | secret 1,2,3,4 | 41.86 | 2.059 |

| | Metric | SNR ↑ | MAE ↓ |
|---|---|---|---|
| 3 audios | cover | 41.45 | 0.232 |
| | secret 1,2,3 | 17.98 | 2.222 |
| 4 audios | cover | 42.56 | 0.153 |
| | secret 1,2,3,4 | 14.81 | 2.860 |

Table 9: Distortion performance of hiding multiple images ($> 2$) into one when the secret INR hidden layer width (D) is set to 86 and 64 for hiding 3 images and 4 images respectively.

Table 10: Distortion performance of hiding multiple audios ($> 2$) into one when the secret INR hidden layer width (D) is set to 86 and 64 for hiding 3 audios and 4 audios respectively.

Additionally, in Table 8, we conduct additional experiments for multi-image steganography hiding more than two images using the same INR structure for the secret data. It shows better performance for the cover/stego pair than Table 6 as the neural network capacity of the stego INR increases as the number of secret data increases. Moreover, as INRSteg does not limit the number of hidden data, the amount of secret data can be set as desired. However, as there is a limit to memory, one may have to decrease the network size of each hidden data in order to fit the memory capacity as in Table 9 and Table 10. The results show that the cover/stego performance is similar to Table 1 as the stego network size width is alike, but the secret/revealed secret performance drops as the secret network size is reduced.

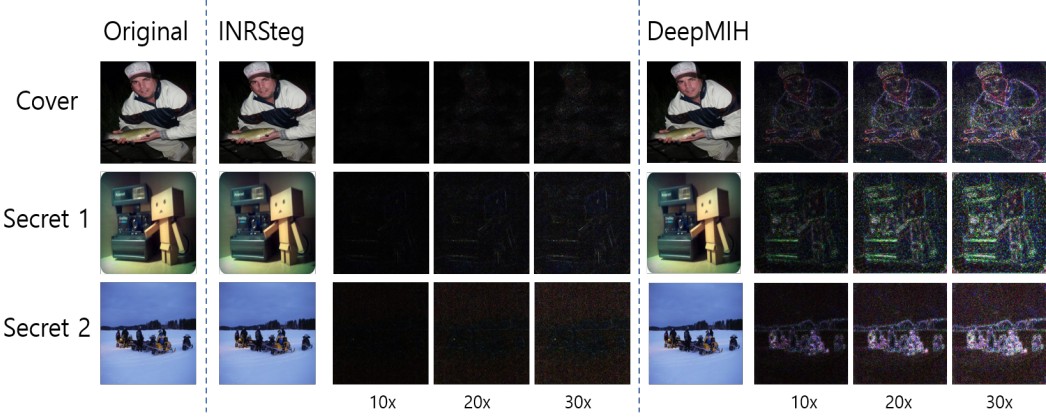

Figure 6: A comparison of difference maps (enhanced by 10x, 20x, and 30x) between the original images and the revealed images. Columns 2-6 are results of INRSteg and columns 7-10 are results of DeepMIH Guan et al. (2023).

To examine security, we adopt two image steganalysis tools, SiaStegNet You et al. (2021) and XuNet Xu (2017). Table 5 and Table 7 presents the detection accuracy of our framework and other models. The security is higher as the detection accuracy is closer to 50%. Unlike other steganography models, the detection accuracy of INRSteg is 50%, meaning that the steganalysis tool completely fails to distinguish the stego data from the cover data. This shows that our framework is undetectable using existing steganalysis tools and assures perfect security. This is because our framework does not directly edit the data in the discrete representation state, but performs manipulation after transforming the representation into a neural network, which is unnoticeable when re-transformed to the discrete representation.

## 4.4 PERMUTATION

Implicit Neural Representations, unlike discrete representations, is not definite, so the distribution of the weight space differs for each data regardless of the modality and the appearance. Therefore, the weight space of the stego data contains borderlines that indicate the positions of the hidden data. To overcome this problem, we propose permutation via private key to ensure security even in the aspect of INRs. After the neural network of the stego data goes through permutation, the weight space becomes indistinguishable as in Figure 7, therefore preventing potential steganalysis algorithms from detecting the existence of secret data. Additionally, through permutation, we can effectively encode the location of the secret data, so that the secret data cannot be revealed without the private key used for permutation.

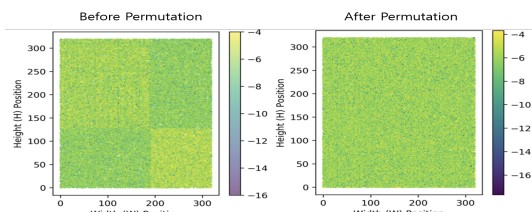

Figure 7: Visualization of the weight distribution of a stego INR before permutation (left) and after permutation(right). Natural logarithm is applied to magnify the weight difference.

## 5 CONCLUSION

In this paper, we propose a novel framework INRSteg, which is capable of hiding multiple cross-modal data within the weight space of Implicit Neural Representations (INR). Our method shows significant improvement in distortion evaluation, capacity, and security in various experiments, including intra and cross-modal steganography, compared to previous steganography methods. For future work, we plan to improve the INR network structure more efficiently and explore the robustness of our framework.

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

## A  TERMINOLOGY

Table 11 presents the notation and the corresponding explanations of key terms used in this paper.

| Notation | Description |
|---|---|
| Overall terminology | |
| $N$ | The number of secret data to be hidden in cover data |
| $x_{secret}$ | Secret data to be hidden |
| $x_{cover}$ | Cover data to be used to fit $\theta_{stego\,prep}$ |
| $x_{stego}$ | Reconstructed data from $\theta_{stego}$ |
| $\theta_{secret}$ | INR fitted on $x_{secret}$ |
| $\theta_{stego\,prep}$ | INR to be fitted on $x_{cover}$ |
| $\theta_{stego}$ | INR fitted on $x_{cover}$ representing $x_{stego}$ |
| $P$ | Permutation matrix for permutation encoding |
| $W$ | Weight vector of INR |
| $b$ | Bias vector of INR |
| INR structure hyperparameter | |
| $n$ | The number of hidden layers |
| $D$ | The number of nodes within a hidden layer |
| $I$ | Input dimension of INR |
| $O$ | Output dimension of INR |

Table 11: Notations in this paper

## B  PSEUDO-CODE ALGORITHM

Algorithm 1 shows the overall hiding steps of our approach, and Algorithm 2 shows the revealing process. Algorithm 3 shows the permutation encoding process.

---

**Algorithm 1** INRSteg

---

**Input** a cover data $x_{cover}$ and N secret data $x_1, x_2,..., x_N$ with private key $\rho$.
**Output** Stego INR $\theta_{encoded\,stego}$

   Initialize $\theta_1, \theta_2, ..., \theta_N$
   **for** $i \leftarrow 1$ to $N$ **do**
      Fit $\theta_i$ to $x_i$
   $\theta_{stego\,prep} \leftarrow \textbf{Concat}(\theta_1, \theta_2, ..., \theta_N)$ with Zeropad(.)
   **for** $i \leftarrow 1$ to $N$ **do**
      Ensure $\theta_i.grad = false$
   Initialize $\theta_{stego\,prep}$
   $\theta_{stego} \leftarrow$ Fit $\theta_{stego\,prep}$ to $x_{cover}$
   $\theta_{encoded\,stego} \leftarrow$ INRPerm($\theta_{stego}; \rho$)
   **return** $\theta_{encoded\,stego}$

---

**Algorithm 2** INRSteg - Retrieve

---

**Input** Stego INR $\theta_{encoded\,stego}$ with private key $\rho$.
**Output** Secret data $x_1, x_2,..., x_N$

   $\theta_{stego} \leftarrow$ InverseINRPerm($\theta_{encoded\,stego}; \rho$)
   **for** $i \leftarrow 0$ to $N - 1$ **do**
      $\theta_i \leftarrow \theta_{stego}[i : i + 1]$
      $x_i \leftarrow$ Reconstruct($\theta_i$)
   **return** $x_1, x_2,..., x_N$

---

---

**Algorithm 3** INRPerm

---

**Input** INR $\theta_{stego}$ with private key $\rho$.
**Output** Permuted INR $\theta_{encoded\,stego}$

   $n \leftarrow$ the number of hidden layers
   **for** $j \leftarrow 1 \,\text{to}\, n - 1$ **do**
      $W \leftarrow$ weight of $j$th hidden layer of $\theta_{encoded\,stego}$
      $b \leftarrow$ bias of $j$th hidden layer $\theta_{encoded\,stego}$
      Permutation matrix $P_j$ derived by $\rho$
      $W^{(j)} = P_j W^{(j)}$
      $W^{(j+1)} = W^{(j+1)} P_j^T$
      $b^{(j)} = P_j b^{(j)}$

   $\theta_{encoded\,stego} \leftarrow$ updated $\theta_{stego}$
   **return** $\theta_{encoded\,stego}$

---

## C   3D Visualization Limitation

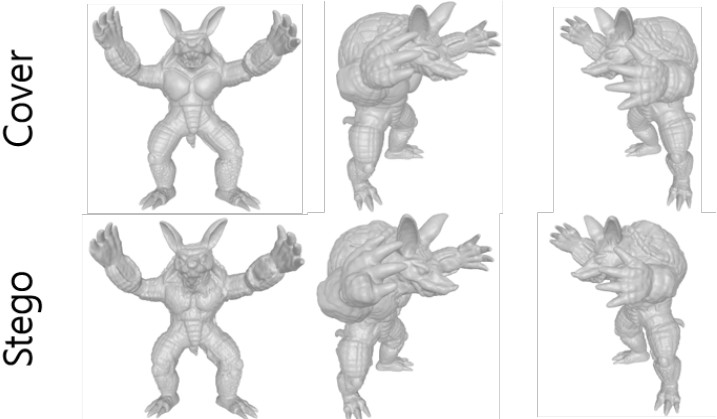

Figure 8: Example of when the stego 3D shape that is not perfectly recovered.

As mentioned in the main paper, due to the underlying unstableness of SIREN Sitzmann et al. (2020), the base structure employed for Implicit Neural Representation (INR), the stego INR may carry defects when the cover modality is 3D shapes as in Figure 8. This limitation can be resolved by adding additional layers to the stego INR or by employing other advanced structures for INRs.

## D   Ablation study: Effectiveness of Padding rate

The ablation study is performed to figure out the effectiveness of the padding rate in single-data steganography. Figure 9 shows the comparison between cover data and stego data which is measured by various evaluation metrics. As shown in Figure 9, as the padding rate is bigger, which means that the space to be fitted on cover data is larger, the network overfits well on cover data, resulting in stego data with high similarities with the cover data. Interestingly, it can be seen that only adding 50% of the network is enough to get high PSNR, which means our method is very efficient in making stego INR. This may be due to the good weight initialization from secret data. Moreover, if the secret and cover data have the same modality, fitting is well performed.

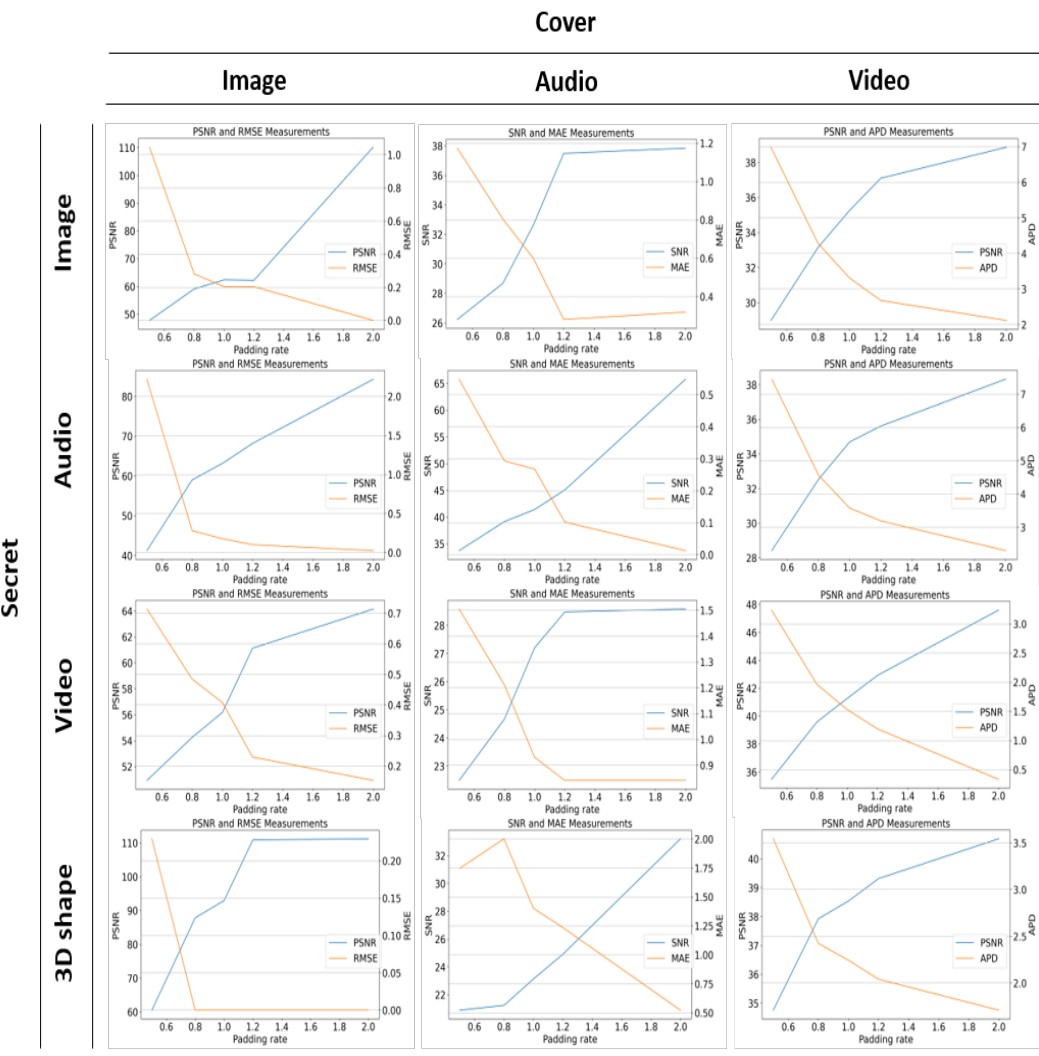

Figure 9: Ablation experiments for hiding single-data steganography varying padding rate. Padding rate refers to the division of padded amount on secret INR by secret INR's hidden layer size. The padding rate varies across $[0.5, 0.8, 1.0, 1.2, 2.0]$. The evaluation metrics are PSNR/RMSE, SNR/MAE, and PSNR/APD for image, audio, and video, respectively. Early stopping is executed in this experiment.

