# OpenReview forum: "INRSTEG: FLEXIBLE CROSS-MODAL LARGE CAPACITY STEGANOGRAPHY VIA IMPLICIT REPRESENTATIONS"
_ICLR.cc/2024/Conference — Submitted to ICLR 2024_

### Official Review · Reviewer_7Brn · 2023-10-30

**Soundness:** 3 good
**Presentation:** 3 good
**Contribution:** 3 good
**Rating:** 5
**Confidence:** 3

**Summary:**

This paper proposes cross-modal high-capacity steganography based on INRs. It occupies part of the weights of the stego INR with the INR containing the secret message and freezes it, and then uses the remaining weights of the stego INR to simulate the function of the cover INR, so as to hide the INR of the secret message while guaranteeing that the function of the stego INR is similar to that of the cover INR.

**Strengths:**

This article proposes a novel INR-based multimodal steganography framework.

**Weaknesses:**

1. Steganography pursues behavioral security, but the framework causes the size of the stego INR to be larger than the size of the normal cover INR, and an attacker may be able to detect the existence of INR steganography based on this anomalous behavior.

2. Security experiments: although this paper can resist traditional image steganalysis, considering that it is similar to neural network steganography, it should be supplemented with experiments on resisting neural network steganalysis.

3. Comparison experiments: Considering that multimodal data can be converted into binary streams, this paper should be supplemented with comparisons with binary stream steganography (e.g., chatgan, etc.).

**Questions:**

How robust is the framework? Can it resist network fine-tuning, pruning, and other operations?

---

> ### Author Response · Authors · 2023-11-13
>
> Thank you for your constructive feedback on our paper. We appreciate the effort you dedicated to providing valuable insights and would like to address each of the comments and suggestions.
>
> **[W1]** Steganography pursues behavioral security, but the framework causes the size of the stego INR to be larger than the size of the normal cover INR, and an attacker may be able to detect the existence of INR steganography based on this anomalous behavior
>
> **[RW1]** We acknowledge that the size of stego INR compared to the size of normal cover INR may provide a potential indicator to attackers, as the number of secret data increases. However, it is important to note that, as mentioned in the paper, we can customize the size of secret INRs. We can fix the stego INR size by setting the size of secret INRs accordingly. Additional experiments regarding this scenario can help the understanding of our framework. Therefore in section 4.3, we will add additional experiments for image and audio where the stego INR size is fixed and the secret INR size decreases as the number of secret data increases.
>
> **[W2]** Security experiments: although this paper can resist traditional image steganalysis, considering that it is similar to neural network steganography, it should be supplemented with experiments on resisting neural network steganalysis.
>
> **[RW2]** In response to your concern, we would like to highlight that, at present, there are no established neural network steganalysis methods available. However, recognizing the importance of addressing potential vulnerabilities, we performed permutation operations and examined the weight distribution plots. The experiment aimed to demonstrate our framework's capability to disrupt potential neural network steganalysis. We acknowledge that developing a neural network steganalysis framework against our work could be an intriguing avenue for future research. We appreciate your suggestion and will consider it for future extension.
>
> **[W3]** Comparison experiments: Considering that multimodal data can be converted into binary streams, this paper should be supplemented with comparisons with binary stream steganography (e.g., chatgan, etc.).
>
> **[RW3]** In response, we would like to emphasize that our chosen baseline for comparison, DeepMIH, has been demonstrated to outperform the previous binary stream steganography methods while achieving larger capacity. Our method exhibits superior performance compared to DeepMIH, validating the efficacy of our proposed method over binary stream steganography. Regarding the relevance of ChatGAN and other binary stream steganography methods to this field, we will enhance our paper by incorporating a comprehensive overview of these methods, including ChatGAN, and their contributions to the related work section.
>
> **[Q]** How robust is the framework? Can it resist network fine-tuning, pruning, and other operations?
>
> **[A]** Our primary focus is to investigate in capacity, distortion, and security of cross-modal steganography, which achieves lossless retrieval of secret information. Therefore, as mentioned in the conclusion section, robustness will be studied for future work.
>
> If there is anything else to discuss, please feel free to let us know.

---

### Official Review · Reviewer_9yJK · 2023-10-31

**Soundness:** 2 fair
**Presentation:** 1 poor
**Contribution:** 2 fair
**Rating:** 3
**Confidence:** 5

**Summary:**

This paper proposes an innovative lossless cross-modal steganography framework based on implicit neural representations (INR). Extensive experiments demonstrate the superiority of the proposed method.

**Strengths:**

N/A

**Weaknesses:**

N/A

**Questions:**

(1)	The English writing should be improved to make this paper readable;
(2)	Is the meaning of “Cross-modal” same with that of “modal-agnostic”? This question should be explained comprehensively.
(3)	I cannot clearly understand details of the proposed framework. I confused how to implement the lossless steganography.
(4)	The experimental results do not evaluate the embedding capacity of the proposed method. I don’t know what is the key reason why the proposed steganography can achieve large capacity.
(5)	Authors merely conduct image steganalysis to evaluate the security of the proposed method, which is not enough for the proposed “modal-agnostic” method.

---

> ### Author Response · Authors · 2023-11-13
>
> Thank you for taking time to review our paper and your comments. We appreciate your feedback and would like to answer to your questions.
>
> **[Q1]** The English writing should be improved to make this paper readable.
>
> **[A1]** We will enhance the English writing to improve the paper's readability as suggested.
>
> **[Q2]** Is the meaning of “Cross-modal” same with that of “modal-agnostic”? This question should be explained comprehensively.
>
> **[A2]** The meaning of “cross-modal” and “modality-agnostic” is different. While both terms are employed in the paper, they are used in different situations. The term “modality-agnostic” refers to the INR itself. We leverage the modality-agnostic nature of INR to propose a steganography framework enabling “cross-modal” tasks. This approach is motivated by the challenges posed by cross-modal tasks in the steganography field. If there are any confusing points, please feel free to let us know what we need to improve.
>
> **[Q3]** I cannot clearly understand details of the proposed framework. I confused how to implement the lossless steganography.
>
> **[A3]** Although there is little loss during the representation transformation phase from discrete representations to INRs, our lossless steganography is achieved in terms of INRs as mentioned in the paper. Once the secret data is transformed into an INR, by placing secret INRs into the stego prep-INR and freezing it during training on the cover data, all weights of secret INRs remain intact throughout the process. Therefore the secret INR is retrieved without any loss.
>
> **[Q4]** The experimental results do not evaluate the embedding capacity of the proposed method. I don’t know what is the key reason why the proposed steganography can achieve large capacity.
>
> **[A4]** Our framework achieves large capacity in two aspects. First, it enables cross-modal tasks even when the dimension or capacity of the secret data exceeds that of the cover data as described in the paper. In section 4.2, there are comprehensive experiments such as hiding 3D shapes in audio. Second, we can hide as much secret data as the sender desires. In the method section, the method of hiding two secret data is explained without loss of generality because multiple data (≥2) can be concealed in the same way using the described method. In section 4.3, we presented experiments of hiding three and four images within a single image. We will add hiding three and four audios within a single audio for clarification.
>
> **[Q5]** Authors merely conduct image steganalysis to evaluate the security of the proposed method, which is not enough for the proposed “modal-agnostic” method.
>
> **[A5]** Our framework conceals data within the INR, therefore introducing no perceptible distortion to the reconstructed data in terms of steganography. Accordingly, existing steganalysis tools are ineffective in detecting the stego data. Even though the steganalysis test is only executed on images in the paper, we believe this is sufficient as the fundamental logic for being undetected operates identically for all modalities and thus the security extends across all modalities.
>
> Thank you for bringing attention to various aspect, and we are open to any further suggestions or insights.

---

### Official Review · Reviewer_AyWV · 2023-11-01

**Soundness:** 3 good
**Presentation:** 3 good
**Contribution:** 2 fair
**Rating:** 5
**Confidence:** 4

**Summary:**

This paper introduces a steganography framework for data represented as Implicit Neural Representations (INR). The method works as follows: multiple secret data are encoded with neural representations, the representations are concatenated without overlap and padded, the padded weights are treated as the only trainable weights and are trained to learn a neural representation for the cover data. Every neural representation is learned with a MLP. The weights in each layer of the final model can be optionally permuted by making use of a private key.

**Strengths:**

The strengths are as follows:
* The authors empirically show that you can hide different modality secret data in different modality cover data.
* The stego data and the recovered secret data both have low distortion.
* Interesting analysis of the weight distribution in section 4.4.

**Weaknesses:**

The weaknesses are as follows:
* The quality of the steganography methods is measured by how much information can be stored in how much space. For instance, image steganography methods report bits per pixel to state how many bits of information can be hidden in each pixel. Similarly, it would be worthwhile to know what the size of the cover data, the secret data is, and the INRs are. It is also a limitation that the INR can be quite large.
* There are multiple missing baselines like SteganoGAN [1] and LISO [2].
* The motivation for hiding data in INRs is not very clear.

[1] Zhang, K. A., Cuesta-Infante, A., Xu, L., & Veeramachaneni, K. (2019). SteganoGAN: High capacity image steganography with GANs. arXiv preprint arXiv:1901.03892.
[2] Chen, X., Kishore, V., & Weinberger, K. Q. (2022, September). Learning Iterative Neural Optimizers for Image Steganography. In The Eleventh International Conference on Learning Representations.

**Questions:**

* Why is hiding audio in audio worse than hiding audio in images?
* Did you train SiaStegNet and XuNet or use a pre-trained model? Do these models operate on INRs or the recovered cover data from the INR?

**Details Of Ethics Concerns:**

No ethics review required.

---

> ### Author Response · Authors · 2023-11-13
>
> Thank you for the detailed feedback. We provide answers to your queries.
>
> 1. Capacity (first weakness)
>
> The sizes of the cover data, secret data, and INRs are detailed in Section 4.1 Experimental Setting. In our experiments, we set the width of the secret data INR to 128 and 192, resulting in a stego INR width between 256 and 384, which is acceptable without suspicions. Additionally, as mentioned in the paper, we can customize the size of secret INRs. Therefore, if there is a fixed stego INR size, we can set the size of the secret INRs accordingly. We acknowledge that additional experiments regarding this scenario can help the understanding of our framework. Therefore in section 4.3, we will add additional experiments in the revision for image and audio where the stego INR size is fixed and the secret INR size decreases as the number of secret data increases.
>
> 2. Missing baselines (second weakness)
>
> We appreciate your comment regarding the missing baselines, SteganoGAN [1] and LISO [2]. These baselines primarily focus on concealing binary messages in images, a different objective compared to our work, which involves modalities such as image, audio, video, and 3D shapes transferable into INRs. While these baselines are excluded from direct comparisons due to this difference, we recognize their impact in the steganography field. Therefore, we will include SteganoGAN and LISO in the related works section to provide a more comprehensive overview.
> Also, one of our chosen baseline, DeepMIH, has demonstrated superiority over previous binary stream steganography methods, which implies the superior performance of our proposed method over those binary stream steganography.
>
> 3. Motivation for using INRs (third weakness)
>
> The motivation behind using INRs lies in their versatility, allowing us to adopt all modalities that can be transferred into INRs for both secret and cover data. This leads to a significant breakthrough in terms of data modality for steganography, which is thoroughly investigated in Section 4.2 Cross-modal Steganography.
>
> 4. Why is hiding audio in audio worse than hiding audio in images? (first question)
>
> For the left table in Table 1, the cover/stego pair performance comparison should be executed within the same cover data modality as different data modalities cannot be compared in terms of performance. Within the same cover modality, we have highlighted the best-performing secret data modality as the secret data modality affects the cover/stego performance, which is explained in 4.2.
> The right table in Table 1 illustrates the secret/revealed secret pair performance. The secret/revealed secret performance remains the same for all experiments as our framework executes lossless secret INR retrieval. This table further provides insight into the representation performance of the INR for each modality.
> Hiding audio in audio and hiding audio in image therefore cannot be compared as they have different cover modalities. The secret/revealed secret performance of audio is the same as explained above. We believe the confusion was caused because the two tables were both placed under “Table 1”. Sorry for this confusion, and we will make this clear in our revision.
>
> 5. Steganalysis experiment settings (second question)
>
> We used a pre-trained model for both steganalysis tools, we are sorry that this wasn’t mentioned in the paper and we will mention this in the revised version. These steganalysis tools do not operate on INRs but are employed as there aren’t any existing steganalysis tools that operate on INRs. Therefore we evaluate them using the recovered cover data (stego data) from the stego INR. As our hiding framework on INRs does not reflect any distortions that indicate steganography in the reconstructed data, existing steganalysis tools are powerless.
> Additionally, recognizing the importance of addressing potential INR steganalysis tools, we perform permutation operations and analyze the weight distribution plots. This experiment aims to demonstrate our framework's capability to disrupt potential neural network steganalysis.

---

### Official Review · Reviewer_bYYf · 2023-11-01

**Soundness:** 4 excellent
**Presentation:** 3 good
**Contribution:** 3 good
**Rating:** 5
**Confidence:** 4

**Summary:**

This paper proposes a framework to hide secret Implicit Neural Representations (INRs) into a cover INR, namely INRSteg.

INRSteg is capable of hiding multiple cross-modal data within the weight space of INR by concatenating multiple secret INRs and permutating the weights INR. When recovering the permutation, the secret INR can be retrieved. INRSteg shows significant improvement in distortion evaluation, capacity, and security in various experiments, including intra and cross-modal steganography, compared to previous steganography methods.

**Strengths:**

originality: the proposed steganography method for INR is simple and novel.

quality: the paper is technically sound.

clarity: the paper is well-organized.

significance: the paper is somehow significant. This paper proposes a way to hide secret INRs in a cover INR, which is not only beneficial for hiding secret data but also for watermarking and copyright protection.

**Weaknesses:**

This paper describes how to encode secret INRs into a cover INR. However, this paper did not do a lot of protection/robustness analysis. For instance, what if the cover INR is being pruned during transmission?

The cover INR would be very big if a lot of secret INRs were embedded. This may be suspicious to malicious attackers who wish to analyze this suspiciously big INR. Although after permutation the weight distribution seems nothing, how about we just sort the weights? The permutation of the weights will not affect the intermediate activations and the outputs, once sorting the weights, will there be any obvious changes? In summary, this paper lacks attack analysis, such as assuming the knowledge of attackers and how attackers can attack (to prevent the final owner from obtaining the correct secret data)/steal (to extract the secret).

**Questions:**

1. can you summarize the advantages/disadvantages of StegaNeRF [1] over the proposed method? Seems like StegaNeRF is also quite related to this proposed method and is one of the latest works in INR steganography.
2. What if the cover INR is being pruned during transmission? Can the secret INR still able to be retrieved?
3. I would be more interested in how robust this proposed method is against the attackers. Start from simple attacks like pruning/noises to the effort trying to reconstruct/retrieve the secret INRs.

[1] https://arxiv.org/abs/2212.01602

---

> ### Author Response · Authors · 2023-11-13
>
> Thank you for your thorough comments. We have addressed them one by one for clarification.
>
> 1. Robustness (first weakness and Q2)
>
> If the cover INR is pruned during transmission, the retrieval of the secret INR is still possible since pruning does not change the location of the weights.  However, the quality of the secret data may be diminished, as the weights of the secret INR will also be affected by the pruning process. We have mentioned robustness as a future work in the conclusion section of the paper. In this paper, we focus on capacity, distortion, and security of a cross-modal steganography framework designed for the lossless retrieval of secret INRs and to block existing steganalysis tools.
>
>
> 2. Big stego INR (second weakness)
>
>  We acknowledge that the size of the stego INR compared to the size of a normal cover INR may provide a potential indicator to attackers, as the number of secret data increases. However, it is important to note that, as mentioned in the paper, we can customize the size of secret INRs. We can fix the stego INR size by setting the size of secret INRs accordingly. Additional experiments regarding this scenario can help the understanding of our framework. Therefore in section 4.3, we will add additional experiments for image and audio where the stego INR size is fixed and the secret INR size decreases as the number of secret data increases.
>
>
> 3. Weight distribution (third weakness)
>
> As you mentioned, the permutation of weights secures the existence of secret data while not affecting the performance of the stego INR. Regarding your inquiry about sorting the weights, we are not sure what you mean by ‘once sorting the weights, will there be any obvious changes?’. Could you please clarify that question?
>
>
> 4. Attack analysis (fourth weakness, Q2, and Q3)
>
>  In the paper, we address potential risks regarding the visibility of the secret data in the weight space. To mitigate this, we employ permutation to eliminate boundaries in the weight space, thus concealing the presence of the secret data. Even though we assume that the attacker is aware that secret data are hidden using permutation, without the private key, the attacker has to employ brute-force guessing to locate the secret data. This is practically impossible when considering the number of possible variations explained in section 3.4. While our paper does not explicitly cover robustness analysis including attacks, we acknowledge this as a valuable extension and have suggested robustness as a future work. Future extensions of our work will cover robustness analysis including various attack scenarios, including those you've suggested.
>
>
> 5. StegaNeRF (Q1)
>
> We realize that StegaNeRF was not mentioned in our paper despite its relevance. We were unaware of StegaNeRF during the writing process, but we will make sure to include a reference to StegaNeRF in our revisions.
> StegaNeRF employs a gradient-based optimization method, introducing a performance trade-off between the secret/revealed secret pair and the cover/stego pair, particularly as the number of hidden data increases. In contrast, INRSteg overcomes this trade-off issue, ensuring high quality for both pairs, regardless of the number of hidden data.
> Additionally, StegaNeRF requires a modal-specific detector for each modality and a classifier, whereas our framework does not require for training of any additional networks. Therefore, INRSteg is free from network bias and is also computationally efficient as any modality can be utilized without requiring a modal-specific network.
> Lastly, utilizing INRs allows our framework to accommodate all modalities that can be transferred into INRs for both secret and cover data. In contrast, StegaNeRF limits the cover data to NeRF and the secret data must be expressible as an ID vector.
> However, our work has yet to explore the robustness of the framework. We leave this for future extensions as our paper focuses on other steganography performance measures which are capacity, distortion, and security.

---

### Meta-Review · Area_Chair_P18Q · 2023-12-05

**Metareview:**

This paper proposed INRSteg, an steganography framework that could hide multiple data without altering the original INR ensuring high-quality stego data. The framework can cross-modal steganography for various modalities including image, audio, video and 3D shapes. As mentioned by reviewers the strengths of this paper are: 1) the proposed steganography method for INR is novel. Weaknesses are: 1) lack of  robustness analysis; 2) writing of the paper could be improved.

Before rebuttal, the ratings from 4 reviewers are 3 "5: marginally below the acceptance threshold" and 1 " 3: reject, not good enough". After rebuttal, 3 reviewers replied and mentioned that their concerns are not addressed and would keep their ratings.

**Justification For Why Not Higher Score:**

reviewers' concerns are not addressed after rebuttal.

**Justification For Why Not Lower Score:**

NA

---

### Decision · Program_Chairs · 2024-01-16

Reject